# ProFD: Prompt-Guided Feature Disentangling for Occluded Person Re-Identification

### Can Cui*
Westlake University
Hangzhou, China
cuican@westlake.edu.cn

### Siteng Huang*
Westlake University
Hangzhou, China
huangsiteng@westlake.edu.cn

### Wenxuan Song
Monash University
Suzhou, China
songwenxuan0115@gmail.com

### Pengxiang Ding
Westlake University
Hangzhou, China
dingpengxiang@westlak.edu.cn

### Min Zhang
Westlake University
Hangzhou, China
zhangmin@westlake.edu.cn

### Donglin Wang†
Westlake University
Hangzhou, China
wangdonglin@westlake.edu.cn

## Abstract

To address the occlusion issues in person Re-Identification (ReID) tasks, many methods have been proposed to extract part features by introducing external spatial information. However, due to **missing part appearance information caused by occlusion** and **noisy spatial information from external model**, these purely vision-based approaches fail to correctly learn the features of human body parts from limited training data and struggle in accurately locating body parts, ultimately leading to misaligned part features. To tackle these challenges, we propose a **Pro**mpt-guided **F**eature **D**isentangling method (**ProFD**), which leverages the rich pre-trained knowledge in the textual modality facilitate model to generate well-aligned part features. **ProFD** first designs part-specific prompts and utilizes noisy segmentation mask to preliminarily align visual and textual embedding, enabling the textual prompts to have spatial awareness. Furthermore, to alleviate the noise from external masks, **ProFD** adopts a hybrid-attention decoder, ensuring spatial and semantic consistency during the decoding process to minimize noise impact. Additionally, to avoid catastrophic forgetting, we employ a self-distillation strategy, retaining pre-trained knowledge of CLIP to mitigate over-fitting. Evaluation results on the Market1501, DukeMTMC-ReID, Occluded-Duke, Occluded-ReID, and P-DukeMTMC datasets demonstrate that **ProFD** achieves state-of-the-art results.

## CCS Concepts

• **Computing methodologies → Object identification**.

## Keywords

CLIP, Occluded Person Re-identification, Feature Disentangling

---

*Both authors contributed equally to this research.
†Corresponding author.

**ACM Reference Format:**
Can Cui, Siteng Huang, Wenxuan Song, Pengxiang Ding, Min Zhang, and Donglin Wang. 2024. ProFD: Prompt-Guided Feature Disentangling for Occluded Person Re-Identification. In *Proceedings of the 32nd ACM International Conference on Multimedia (MM '24), October 28-November 1, 2024, Melbourne, VIC, Australia.* ACM, New York, NY, USA, 10 pages. https://doi.org/10.1145/3664647.3680958

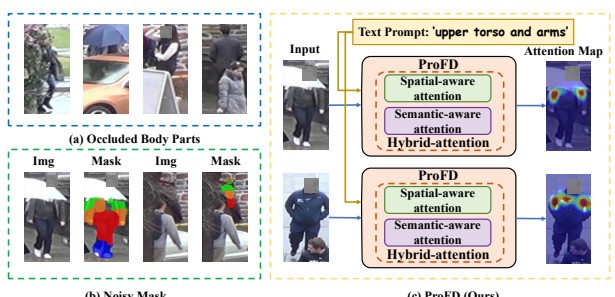

**Figure 1: Two crucial challenges of occluded person ReID. (a) Missing Information caused by occlusion. (b) Noise in external spatial information. (c) Our proposed Prompt-guided Feature Disentangling method (ProFD).**

## 1 Introduction

Person Re-Identification (ReID) refers to finding images in a database that match a given query image. However, in complex urban environments, occlusions between people or between people and objects often occur. These occlusions can lead to severe misalignment, noise, and missing information issues, thereby significantly degrading the identification performance [1]. Therefore, to address this issue, researchers have actively engaged in the task of occluded person re-identification. The existing solution aims to extract well-aligned part features. They can be roughly divided into two categories: external-cue-based methods and attention-based methods. External-cue-based methods [2–11] rely on external cues from off-the-shelf models or additional supervision to provide spatial information for aiding in the locating and alignment of body parts. Attention-based methods [12–20] address misalignment through emphasizing salient regions and suppressing background noise without utilizing external information.

However, these aforementioned purely vision-based approaches still face the following two key problems, depicted in Figure 1 (a) and (b): **(1) Missing part appearance information caused by occlusion**: Occlusion can cause the loss of visual information for certain body parts in the training data, significantly reducing the frequency of these parts' appearance in the dataset. **(2) Noisy spatial information from external model**: Due to the domain gap between the training data of the external model and the ReID datasets, the pseudo-labels generated by the external model inevitably contain errors, introducing noise into the pseudo-labels. As a result, the model struggles to accurately locate part features of the human body, ultimately leading to misaligned part features.

To reduce the impact brought by the missing information and noisy label problems, we propose a **Pro**mpt-guided **F**eature **D**isentangling framework (**ProFD**). By incorporating the rich pre-trained knowledge of textual modality, our framework helps the model accurately capture well-aligned part features of the human body, as shown in Figure 1 (c) . Firstly, we design part-specific prompts for different body parts, which are fed into the text encoder of CLIP to initialize the decoder's query embeddings. In this way, the model can be trained with semantic priors, alleviating the issue of body parts data scarcity and thus improving the model's performance. Additionally, we design an auxiliary segmentation task to aid in the initial spatial-level alignment of text prompts and visual feature maps, enabling the prompts to have some spatial awareness. Then, to mitigate the influence of the noise spatial information, we propose a hybrid-attention decoder to generate well-aligned part features. This decoder contains two types of attention mechanisms: spatial-aware attention and semantic-aware attention. The spatial-aware attention relies on external noisy spatial information to ensure spatial consistency of part features. On the other hand, the semantic-aware attention is derived from the text modality information of the pre-trained CLIP model. Due to the generalizability of semantic information, it can serve as a complement to spatial-aware attention to reduce the impact of noise. Furthermore, to alleviate catastrophic forgetting during fine-tuning, we propose a self-distillation strategy, using memory banks to store the pre-trained knowledge of CLIP and guide the output features during training.

This paper evaluates the efficacy of **ProFD** on two holistic datasets: Market1501 [21] and DukeMTMC-ReID [22], and three occluded datasets: Occluded-Duke [3], Occluded-ReID [23] and P-DukeMTMC [22]. Experimental results demonstrate that **ProFD** performs competitively with previous state-of-the-art methods. Moreover, owing to introduce textual modality and self-distillation strategy, ProFD demonstrates strong generalization capabilities, significantly outperforming other methods on the Occluded-ReID dataset [23], with improvements of at least **8.3%** in mAP and **4.8%** in Rank-1 accuracy.

The key contributions of this paper are threefold:

- We introduce a novel framework **ProFD** to efficiently utilize textual prompts to guide part feature disentangling for occluded person re-identification.
- We propose a new self-distillation strategy for part features to better preserve pre-trained Multi-modal knowledge and alleviate overfitting.

- We conduct extensive experiments on the holistic datasets Market1501 [21] and DukeMTMC-ReID [23], and the occluded datasets Occluded-Duke [3], Occluded-ReID [23] and P-Duke-MTMC [22], which demonstrate that our method surpasses lot of previous methods and sets state-of-the-art.

## 2 Related Work

### 2.1 Occluded Person Re-Identification

Compared to holistic person re-identification, occluded person re-identification is more challenging due to information incompleteness and spatial misalignment. To mitigate the spatial misalignment issue, several approaches [24–27] adopt manual partitioning of the input image and utilize part pooling to generate local feature representations. However, hand-crafted cropping is impractical and might introduce subjective bias. To solve those issues, other methods [3, 4, 28] utilize additional information for the localization of human body parts, such as segmentation, pose estimation, or body parsing. They leverage these auxiliary information in both training and test phases. However, others [9, 10] only utilize that extra clues to guide the learning process.

Currently, attention-based methods [12–19, 29] have gained considerable interest from the ReID community, primarily driven by the powerful feature extraction and disentangling capabilities of transformers. He et al. [14] introduce TransReID, a transformer framework, demonstrating its remarkable feature extraction capabilities through experiments. Li et al. [29] pioneer the Part Aware Transformer (PAT) for occluded person ReID, showcasing its effectiveness in robust human part disentanglement. While the aforementioned methods partially address the occlusion issue, they all predominantly focus on visual modality to struggle with the challenges brought by the missing information and noisy spatial information.

### 2.2 Vision-Language Learning

Vision-language models encompass diverse categories [30–35] in the face of different research queries. In this work, we mainly focus on the representation models, which aim to learn common embeddings for both images and texts. The idea of cross-modality alignment is not new and has been studied with drastically different technologies [36–39]. Recently, with the huge advancements in vision-language pretrained model, the concurrent learning of image and text encoders has been a notable development [40, 41] An exemplary contribution in this domain is the contrastive vision-language pre-training framework [42–44], denoted as CLIP [45], which facilitates effective few-shot or even zero-shot classification [46, 47] by pre-trained on 400 million text-image pairs.

Despite the considerable headway in CLIP, the effective adaptation of these pre-trained models to downstream tasks remains a formidable challenge. Noteworthy endeavors in this realm include Context Optimization (CoOp) [48] and Conditioned Context Optimization (CoCoOp) [49], which employ learnable text embeddings to assist with image classification. Similarly, CLIP-adaptor [50] and TIPadaptor [51] utilize lightweight adaptors to better fine-tune on few-shot downstream tasks with little trained parameters. Dense-CLIP [52] introduces a language-guided fine-tuning approach for semantic and instance segmentation tasks instead of image classification. The study by GLIP [53] delves into the deep fusion of

semantic-rich image-text pairs to attain a unified formulation for object detection and phrase grounding. Our approach leverages CLIP to inject rich textual knowledge into occluded person ReID task, aiding in the generation of well-aligned part features.

## 3 Preliminary

***Contrastive language-image pre-training.*** CLIP comprises two encoders—a visual encoder (typically ViT [54] or ResNet [55]) and a text encoder (typically Transformer [56]). The objective of CLIP is to align the embedding spaces of visual and language modalities. And CLIP can be used for zero-shot classification in aligned embedding space. The text is obtained by a predefined template, such as "a photo of a $\{class_i\}$.", where $\{class_i\}$ represents the $i$-th class name. This input text is then fed into the text encoder to generate $\{w_i\}_{i=1}^{K}$, a set of weight vectors, each representing different category (a total of $K$ categories). Simultaneously, image features $x$ are generated by the image encoder. Then, compute similarities between the image vectors and the text vectors, followed by a softmax operation to derive prediction probabilities, which is formulated as:

$$p(y|x) = \frac{\exp(\text{sim}(x, w_y)/\tau)}{\sum_{i=1}^{K} \exp(\text{sim}(x, w_i)/\tau)}, \tag{1}$$

where $\text{sim}(\cdot, \cdot)$ denotes cosine similarity and $\tau$ is a learned temperature parameter.

***Prompt-based learning.*** To enhance the transfer capabilities of the CLIP model and mitigate the need for prompt engineering, the CoOp [48] approach is introduced. Instead of using "a photo of " as the context, CoOp introduces $M$ learnable context vectors, $\{v_1, v_2, \ldots, v_M\}$, each having the same dimension with the word embeddings. The prompt for the $i$-th class, denoted by $T_i$, now becomes:

$$T_i = \{v_1, v_2, \ldots, v_M, c_i\}, \tag{2}$$

where $c_i$ is the word embedding for the class name. The context vectors are shared among all classes. Let $g(\cdot)$ denote the text encoder, the prediction probability is formulated as:

$$p(y|x) = \frac{\exp(\text{sim}(x, g(T_y))/\tau)}{\sum_{i=1}^{K} \exp(\text{sim}(x, g(T_i))/\tau)}. \tag{3}$$

where $\text{sim}(\cdot, \cdot)$ denotes cosine similarity and $\tau$ is a learned temperature parameter. Notably, the base model of CLIP remains frozen throughout the entire training process.

## 4 Methodology

We proposed a CLIP-based framework, named **Pro**mpt-guided **F**eature **D**isentangling (ProFD). The overall framework of ProFD is illustrated in Figure 2. It mainly consists of three components. First, to reduce the effect of missing information, we design several part-specific prompts contained with rich semantic priors from CLIP and utilize external noisy segmentation masks as supervision to pre-align visual-textual modality in spatial level (Sec. 4.1). Second, to alleviate the effect brought by the noisy mask, we propose a hybrid-attention decoder to generate better-aligned part features. (Sec. 4.2). Third, to overcome the catastrophic forgetting problem, we propose a self-distillation strategy to store pre-trained knowledge of CLIP with memory banks. (Sec. 4.3).

## 4.1 Part-aware Knowledge Adaptation

*4.1.1 Part-specific Text Prompts.* To alleviate the missing information problem brought by occlusion, we design a set of Part-specific text prompts to introduce the pre-trained language knowledge of CLIP about human body parts. In contrast to classification tasks, human parsing focuses on identifying the locations of various body regions in each image. Thus, it is difficult to manually design a set of prompts that are optimal for the human parsing task. Moreover, recent studies [48, 57, 58] have shown that learnable templates are more beneficial for adapting to downstream tasks compared to fixed and manually designed templates. Thus, we employ a trainable template comprising $M$ learnable prefix tokens to create a prompt template, as illustrated in Equation 2, which is better suited for human parsing. And we substitute $c_i$ with the labels of distinct human body parts $p_n$, such as 'head','torso','feet',etc., to generate $N$ text prompts, i.e.,

$$T_n = \{v_1, v_2, \ldots, v_M, p_n\}, \tag{4}$$

where $v_i$, $i \in \{1, 2, ..., M\}$ represents the learnable prefix tokens, $p_n$ represents the $n$-th body part name, and $N$ is the number of parts. The $N$ text prompts are mapped to the shared embedding space by using a pre-trained text encoder $E_t$ to get the prompt embedding $E_{pro} = E_t(T_n), E_{pro} \in \mathbb{R}^{N \times d}$.

*4.1.2 Spatial-level Alignment.* Due to the lack of spatial-level alignment between text features and visual feature map $F \in \mathbb{R}^{H \times W \times d}$ during pre-training, it's hard to locate body regions according to textual instruction, as required for subsequent steps. Therefore, we design an auxiliary semantic segmentation task to restore the locality of feature map and realize spatial-level alignment.

Spatial-level alignment aims to establish dense connections between text prompts and image features, enabling text prompts to have spatial awareness. Following normalization, the extracted prompt embeddings and image features are employed to query the presence probability of different regions through inner product computation. Thus, the presence score at each spatial position is achieved as:

$$S_{ij}^n = F_{ij} \cdot E_{pro}^n,$$
$$i = 1, ...., H; j = 1, ..., W; n = 1, ..., N, \tag{5}$$

where $F_{ij} \in \mathbb{R}^{1 \times d}$ is the feature vector of $F$ at pixel $(i, j)$, and $E_{pro}^n$ is the $n$-th prompt embedding in $E_{pro} \in \mathbb{R}^{N \times d}$.

The estimated score map is supervised by the target mask $\mathcal{M} \in \mathbb{R}^{H' \times W' \times N}$, which is obatined with the off-the-shelf model Pifpaf[59], via the following spatial-level alignment loss:

$$L_{align} = \text{CE}(\mathcal{S}, \text{AP}(\mathcal{M})), \tag{6}$$

where $\mathcal{S} = \{S_{ij}^n\}_{n=1}^{N}, \mathcal{S} \in \mathbb{R}^{H \times W \times N}$, $\text{CE}(\cdot)$ is the cross entropy loss, $\text{AP}(\cdot)$ represents the average pooling function used to generate patch labels $\mathcal{M}_p$. Its stride and kernel size are the same as the setting used by patchifying the image.

## 4.2 Prompt-guided Part Feature Disentangling

*4.2.1 Hybrid Attention Decoder.* To reduce the impact of noisy spatial information from off-the-shelf model, we introduce a hybrid attention decoder, which utilizes a set of part-specific prompts $E_{pro}$ as queries to incorporate relevant information into part features

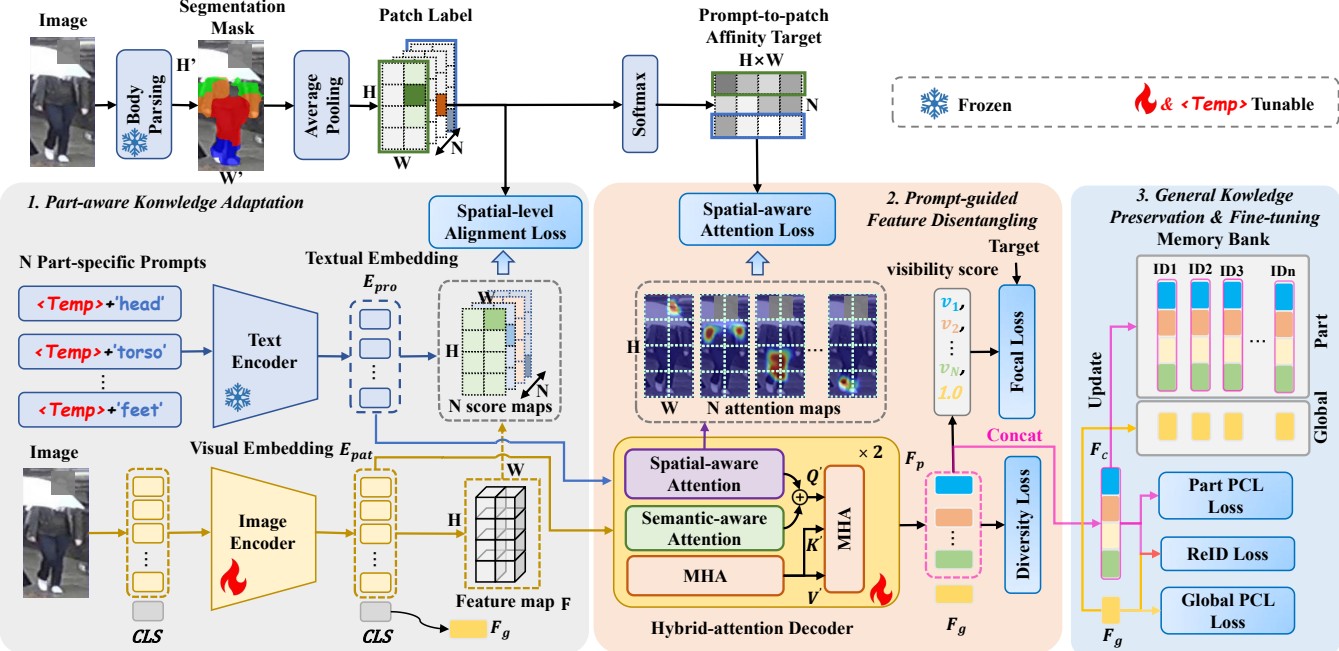

**Figure 2: Illustration of our proposed ProFD framework. It mainly contains three components: (1) Part-aware Knowledge Adaptation(left), (2) Prompt-guided Feature Disentangling(middle), (3) General Knowledge Preservation & Fine-tuning(right). Part-aware Knowledge Adaptation aims to adapt CLIP to Occluded Person ReID task. Prompt-guided Feature Disentangling employ hybrid-attention decoder to extract corresponding part features from holistic feature map based on textual prompt. For a more detailed structure of hybrid-attention, please refer to Figure 3. General Knowledge Preservation utilize global and part memory banks to avoid pre-trained knowledge forgetting of CLIP during fine-tuning.**

$F_p = \{f_p^i\}_{i=1}^N \in \mathbb{R}^{N \times d}$. The hybrid attention decoder consists of multiple hybrid attention blocks with spatial-aware attention and semantic-aware attention, as illustrated in Figure 3.

First, to further strengthen the semantic information of visual features, we use reverse cross attention mechanism to absorb information into patch tokens $E_{pat}$ from $E_{pro}$.

$$E'_{pat} = \text{MHA}(E_{pat}, E_{pro}), \tag{7}$$

where $\text{MHA}(\cdot, \cdot)$ is the standard multi-head attention function, $F'$ represent the updated keys.

Then, feed the queries $E_{pro}$ and keys $E_{pat}$ into hybrid attention module, which includes two kinds of attention. One of them is the spatial-aware attention and the other is semantic-aware attention. The spatial-aware attention branch obtains spatially perceived attention through introducing external mask supervision, specifically as follows:

$$SPA(E_{pro}, E_{pat}) = \text{softmax}(\frac{E_{pro}(E_{pat}W_k)^T}{\sqrt{d}})E_{pat}W_v, \tag{8}$$

where $E_{pro}(E_{pat}W_k)^T \in \mathbb{R}^{N \times HW}$ is the affinity matrix between prompts and patch tokens, $(E_{pat}W_k)$ and $(E_{pat}W_v)$ represents queries and keys, respectively. The $W_{k,v} \in R^{d \times d}$ are linear projection. The ideal cross attention distribution should emphasize body part regions to suppress irrelevant noise, for which we use external coarse and noisy patch labels $\mathcal{M}_p$ to supervise attention.

And the loss function is defined as:

$$L_{attn} = \sum \text{softmax}(\mathcal{M}_p^T)\log(\text{softmax}(\frac{E_{pro}(E_{pat}W_k)^T}{\sqrt{d}})). \tag{9}$$

To mitigate the influence of noise in spatial information on spatial-aware attention, semantic-aware attention relies on the semantic correlation between textual prompts and visual tokens, and it assigns more attention to patch tokens with similar semantics. The specific formula is as follows:

$$SEA(E_{pro}, E_{pat}) = \text{MHA}(E_{pro}, E_{pat}). \tag{10}$$

Finally, the output embedding of these two types of attention are summed and sent into the feed-forward network to obtain the final part features $F_p$, which are as follows:

$$F_p = \text{FNN}(\text{MHA}(E_{pro}^{spa} + E_{pro}^{sea}, E'_{pat})), \tag{11}$$

where $\text{FNN}(\cdot)$ represents the feed-forward network [56], which first maps the feature from dimension $d$ to $4d$ linearly, applies GeLU and Dropout, then maps back to dimension $d$. $E_{pro}^{spa}$ and $E_{pro}^{sea}$ are the outputs of the two attention mechanisms, respectively.

In addition, in order to reduce the redundancy between part features, we apply diversity loss for the part features learning, which is defined as follows:

$$L_{div} = \frac{1}{N(N-1)}\sum_i \sum_j d_{ij}, \quad i < j, \quad i, j = 1, ..., N \tag{12}$$

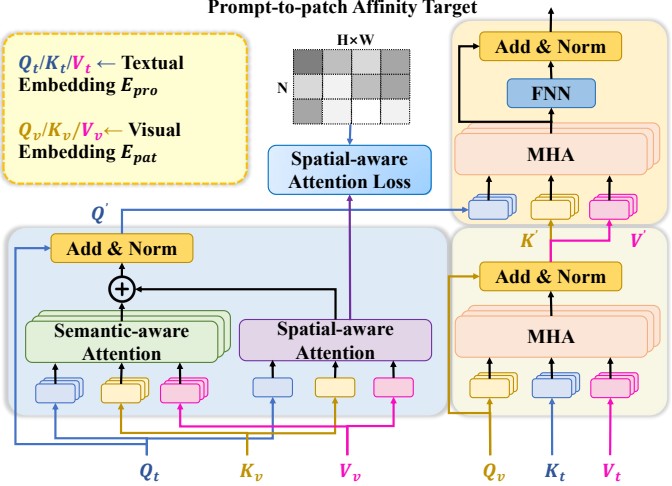

**Figure 3: The architecture of hybrid attention decoder.**

where $d_{ij} = |cos(f_p^i, f_p^j)|$, $f_p^i$ and $f_p^j$ represent the any two part features of $F_p \in \mathbb{R}^{N \times d}$.

*4.2.2 Body Part Visibility Estimation.* To filter out the features of occluded body parts, a visibility score $v_i$ for each feature should be predicted, where 0/1 corresponds to invisible/visible parts, respectively. The visibility scores are only used during the inference phase, to alleviate the issue of feature misalignment. The holistic features have visibility scores set to 1. i.e., $v_g = 1$. For body part-based features, visibility score $v_i$ with $i \in \{1, ..., N\}$ are individually predicted by different binary classifiers. And it is supervised by the focal loss [60], as below formulation:

$$L_{vis} = \begin{cases} -\alpha(1-v_i)^\gamma log(v_i) & \text{if } \hat{v}_i = 1, \\ -(1-\alpha)v_i^\gamma log(1-v_i) & \text{otherwise,} \end{cases} \quad (13)$$

where $\alpha$ and $\gamma$ are hyperparameters that control the balance between positive and negative samples, and $\hat{v}_i$ represents the target visibility of part i, which is assigned a value of 1 if at least one pixel of image is categorized as the i-th region.

In the inference stage, the distance between query and gallery samples is defined as:

$$d = \frac{\sum_{i=1}^{N}(v_i^q \cdot v_i^g)d_n + d_g}{\sum_{i=1}^{N}(v_i^q \cdot v_i^g) + 1}, \quad (14)$$

where $v_i^q$ and $v_i^q$ represent the i-th part visibility score from query and gallery sample, respectively. $d_n$ and $d_g$ indicate cosine distance between the n-th part features and cosine distance between global features, separately.

## 4.3 General Knowledge Preservation

The experimental results of CLIP-ReID [61] indicate that the pretrained knowledge preserved through prompt learning is beneficial for the ReID task. However, the two-stage training process of CLIP-ReID is overly complex and inefficient. Fortunately, it has been demonstrated that prompt learning is not necessary for knowledge preservation of CLIP [62]. Thus, we propose a new single-stage

training strategy for avoiding catastrophic forgetting for the occluded scenarios. Due to the different characteristics of global and part features, we implement knowledge preservation from two perspectives: global and local knowledge preservation.

*4.3.1 Global Knowledge Preservation.* To preserve pre-trained knowledge of global features, an external memory bank $\mathcal{K}_g \in \mathbb{R}^{d \times C}$ is created to hold the feature centroids of all ID classes. Each centroid is initialized by averaging the visual features of all images belonging to that ID. During the fine-tuning of visual encoder, the centroid is updated using momentum according to the following procedure:

$$\mathcal{K}_g[y_i] \leftarrow m_g \mathcal{K}_g[y_i] + (1-m_g)F_g^i, \quad (15)$$

where $y_i$ and $F_g^i$ means the ID label of sample i and its global feature, separately, and $m_g$ represents a momentum factor that governs the speed of updates. The PCL loss [62] is defined as follows:

$$L_{pcl}^g = -\log\frac{\exp(s(\mathcal{K}_g[y_i], F_g^i)/\tau)}{\sum_{j=1}^{C} \exp(s(\mathcal{K}_g[j], F_g^i)/\tau)}, \quad (16)$$

where $S(\cdot, \cdot)$ represents cosine similarity between vectors, $\mathcal{K}_g[j]$ refers to the feature center of class j stored in a memory bank $\mathcal{K}_g$.

*4.3.2 Local Knowledge Preservation.* For part features, their similarity is independent of ID labels, such as different people may have similar part appearances. Due to the lack of annotations for part features, the way to distill knowledge for part features differs from global features.

To solve this problem, we establish a part memory bank $\mathcal{K}_p \in \mathbb{R}^{Nd \times C}$ to store all ID centers of concatenated part features $F_c = [f_p^1; f_p^2; ... f_p^N]$, which can be regard as a type of ID-relevant global feature. And the memory bank $\mathcal{K}_p$ is updated with corresponding concatenated features part $F_c$ and momentum $m_p$. In the initial stages of training, as the random-initialized decoder do not have strong feature disentanglement capability, we utilize external segmentation masks and weighted average pooling to extract part features, used to initialize the memory. This process can be formulated as follows:

$$\text{Init. : } \mathcal{K}_p[y_i] := [\text{WAP}(F, \mathcal{M}_p^1); \text{WAP}(F, \mathcal{M}_p^2); ...; \text{WAP}(F, \mathcal{M}_p^N)]$$
$$\mathcal{K}_p[y_i] \leftarrow m_p \mathcal{K}_p[y_i] + (1-m_p)F_c^i, \quad (17)$$

where $F$ represents the feature map, $\mathcal{M}_p^1, \mathcal{M}_p^2, ..., \mathcal{M}_p^N$ denote patch labels of N body parts. And the other symbols maintain their previously defined meanings. The objective is formulated as:

$$L_{pcl}^p = -\log\frac{\exp(s(\mathcal{K}_p[y_i], F_c^i)/\tau)}{\sum_{j=1}^{C} \exp(s(\mathcal{K}_p[j], F_c^i)/\tau)}. \quad (18)$$

## 4.4 Overall Objective

During traning process, we use cross entropy loss and triplet loss for global and part features. The formulation is as follows:

$$\begin{aligned} L_{total} = & L_{id}(F_g) + L_{tri}(F_g) + L_{id}(F_c) + L_{tri}(F_c) \\ & + L_{div}(F_p) + L_{pcl}^p(F_c) + L_{pcl}^g(F_g) + L_{align} + L_{attn} + L_{vis} \end{aligned} \quad (19)$$

where $L_{id}$ and $L_{tri}$ represent cross entropy loss and triplet loss, respectively, $F_g$ is global feature obtained from the visual encoder

of CLIP, $F_p$ and $F_c$ represents the part features and the concatenate part feature, respectively.

## 5 Experiments

### 5.1 Datasets and Metrics

**Datasets.** To highlight that our model maintains performance on holistic datasets and demonstrates improvement on occluded datasets, we selected the following datasets: holistic datasets, including Market1501 [21] and DukeMTMC-ReID [22], as well as occluded datasets, namely Occluded-Duke [3], Occluded-ReID [23], and P-DukeMTMC [22]. The details are shown as follows:

- **Market1501:** Comprising 32,668 labeled images of 1,501 identities captured by 6 cameras, this dataset is divided into a training set with 12,936 images representing 751 identities, used exclusively for model pre-training.
- **DukeMTMC-ReID:** This dataset consists of 36,411 images showcasing 1,404 identities from 8 camera. It includes 16,522 training images, 17,661 gallery images, and 2,228 queries.
- **Occluded-Duke:** Containing 15,618 training images, 2,210 occluded query images, and 17,661 gallery images, this dataset is a subset of DukeMTMC-ReID, featuring occluded images and excluding some overlapping ones.
- **Occluded-ReID:** Captured by mobile camera equipment on campus, this dataset includes 2,000 annotated images belonging to 200 identities. Each person in the dataset is represented by 5 full-body images and 5 occluded images with various types of occlusions.
- **P-DukeMTMC:** Derived from the DukeMTMC-ReID dataset, this modified version comprises 12,927 images (665 identities) in the training set, 2,163 images (634 identities) for querying, and 9,053 images in the gallery set.

**Evaluation Metrics.** Following established conventions in the ReID community, we assess performance using two standard metrics: the Cumulative Matching Characteristics (CMC) at Rank-1 and the Mean Average Precision (mAP). Evaluations are conducted without employing re-ranking [63] in a single-query setting.

### 5.2 Implementation Details.

**Model Architecture.** We use ViT-based CLIP as our backbone, which contains 12-layer 6-head transformer. As CLIP-ReID [61], we use a linear projection to reduce the extracted feature dimension from 762 to 512. Based on this backbone, it is further extended with a 2-layer 8-head transformer to learn hybrid attention and extract the important part features.

**Training Details.** The training procedure mainly follows the CLIP-ReID's setting [61]. During training and inference process, the input images are resized to $256 \times 128$ and patch size is $16 \times 16$. During the training phase, person images undergo data augmentation techniques including random flipping, random erasing, and random cropping, each applied with a probability of 50%. The batch size is configured as 64, consisting of 4 images per person. The hyper-parameters of focal loss $\alpha$ and $\gamma$ are set to 0.65 and 2. The momentum $m_g$ and $m_p$ are both set to 0.2. The temperature $\tau$ of PCL loss equals to 0.05 The Adam optimizer is utilized with a weight decay factor of 0.0005. The learning rate starts at 5e-5 and is reduced by a factor of

**Table 1: Performance comparison of the occluded ReID problem on the Occluded-Duke, Occluded-ReID and P-DukeMTMC. These previous methods are classified into three groups from top to bottom: holistic feature based, external cues based, and attention based. ∗ indicates the back bone is with an overlapping stride setting, stride size $s_o = 12$.**

| Method | Occluded-Duke | | Occluded-ReID | | P-DukeMTMC | |
|---|---|---|---|---|---|---|
| | Rank-1 | mAP | Rank-1 | mAP | Rank-1 | mAP |
| Part-Aligned [64] | 28.8 | 20.2 | - | - | - | - |
| PCB [65] | 42.6 | 33.7 | 41.3 | 38.9 | - | - |
| Adver Occluded [66] | 44.5 | 32.2 | - | - | - | - |
| CLIP-ReID [61] | 67.1 | 59.5 | - | - | - | - |
| CLIP-ReID* [61] | 67.2 | 60.3 | - | - | - | - |
| PVPM [2] | 47.0 | 37.7 | 70.4 | 61.2 | 51.5 | 29.2 |
| PGFA [3] | 51.4 | 37.3 | - | - | 44.2 | 23.1 |
| HOReID [4] | 55.1 | 43.8 | 80.3 | 70.2 | - | - |
| GASM [5] | - | - | 74.5 | 65.6 | - | - |
| VAN [6] | 62.2 | 46.3 | - | - | - | - |
| OAMN [7] | 62.6 | 46.1 | - | - | - | - |
| PGFL-KD [8] | 63.0 | 54.1 | 80.7 | 70.3 | 81.1 | 64.2 |
| BPBreID [9] | 66.7 | 54.1 | 76.9 | 68.6 | 91.0 | 77.8 |
| RGANet [10] | **71.6** | 62.4 | 86.4 | 80.0 | - | - |
| PAT [12] | 64.5 | 53.6 | 81.6 | 72.1 | - | - |
| DRL-Net [13] | 65.8 | 53.9 | - | - | - | - |
| TransReID [14] | 66.4 | 59.2 | - | - | - | - |
| MHSA [15] | 59.7 | 44.8 | - | - | 70.7 | 41.1 |
| FED [16] | 68.1 | 56.4 | 86.3 | 79.3 | - | - |
| MSDPA [17] | 70.4 | 61.7 | 81.9 | 77.5 | - | - |
| FRT [18] | 70.7 | 61.3 | 80.4 | 71.0 | - | - |
| DPM* [19] | 71.4 | 61.8 | 85.5 | 79.7 | - | - |
| SAP* [11] | 70.0 | 62.2 | 83.0 | 76.8 | - | - |
| **ProFD** (Ours) | 70.8 | 62.8 | 91.1 | 88.5 | 91.7 | 83.7 |
| **ProFD*** (Ours) | 70.6 | **63.1** | **92.3** | **90.3** | **92.8** | **84.7** |

0.1 at the 30th and 50th epochs, respectively. And training process terminates after 120 epochs. During both training and inference, the CLIP text encoder remains frozen. We choose five human body part categories that feed into the text encoder, which include 'head', 'upper arms and torso', 'lower arms and torso', 'legs', and 'feet'. All training and experiments are performed with one NVIDIA V100 GPU.

### 5.3 Comparison with the State-of-the-Art

*5.3.1 Evaluation on Occluded Person ReID Dataset.* To demonstrate the performance of our proposed method, we evaluate our method on three public occluded Person ReID datasets, which specifically consist of Occluded-Duke [3], Occluded-ReID [23], and P-DukeMTMC [22]. The experimental results are shown in Table 1. The recent state-of-the-art methods of ReID can be devided into three groups: holistic feature based method [61, 64–66], external cues based method [2–11] and attention based methods [12–19].

Our method significantly outperforms all other methods, achieving a rank-1 accuracy/mAP of 70.8%/62.8% on Occluded-Duke, 91.1%/88.5% on Occluded-ReID, and 91.7%/83.7% on P-DukeMTMC, respectively. For instance, RGANet also employs CLIP as its backbone and utilizes external segmentation results as supervision to extract aligned part features, which has achieved state-of-the-art

**Table 2: Performance comparison of the holistic ReID problem on the Market1501 and DukeMTMC-ReID. These SOTA methods are divided into two groups from top to bottom: holistic ReID method, Occluded ReID method. ∗ indicates the back bone is with an overlapping stride setting, stride size $s_O = 12$.**

| Method | Market1501 Rank-1 | mAP | DukeMTMC-ReID Rank-1 | mAP |
|---|---|---|---|---|
| MGN [67] | 95.7 | 86.9 | 88.7 | 78.4 |
| PCB [65] | 92.3 | 77.4 | 81.7 | 66.1 |
| PCB+RPP [65] | 93.8 | 81.6 | 83.3 | 69.2 |
| VPM [68] | 93.0 | 80.8 | 83.6 | 72.6 |
| Circle [69] | 94.2 | 84.9 | - | - |
| ISP [70] | 95.3 | 88.6 | 89.6 | 80.0 |
| TransReID [14] | 95.2 | 88.9 | 90.7 | 82.6 |
| DC-Former* [71] | 96.0 | 90.4 | - | - |
| CLIP-ReID [61] | 95.5 | 89.6 | 90.0 | 82.5 |
| CLIP-ReID* [61] | 95.4 | 90.5 | 90.8 | 83.1 |
| PCL-CLIP [62] | 95.9 | **91.4** | - | - |
| PGFA [3] | 91.2 | 76.8 | 82.6 | 65.5 |
| PGFL-KD [8] | 95.3 | 87.2 | 89.6 | 79.5 |
| HOReID [4] | 94.2 | 84.9 | 86.9 | 75.6 |
| MHSA [15] | 94.6 | 84.0 | 87.3 | 73.1 |
| BPBreID [9] | 95.1 | 87.0 | 89.6 | 78.3 |
| RGANet [10] | 95.5 | 89.8 | - | - |
| PAT [12] | 94.2 | 84.9 | 88.8 | 78.2 |
| FED [16] | 95.0 | 86.3 | 89.4 | 78.0 |
| DPM* [19] | 95.5 | 89.7 | 91.0 | 82.6 |
| FRT [18] | 95.5 | 88.1 | 90.5 | 81.7 |
| PFD* [72] | 95.5 | 89.7 | 91.2 | 83.2 |
| SAP* [11] | **96.0** | 90.5 | - | - |
| **ProFD** (Ours) | 95.1 | 90.0 | 91.7 | 83.2 |
| **ProFD*** (Ours) | 95.6 | 90.8 | **92.1** | **84.0** |

**Table 3: Performance of ProFD with different attention mechanism on Occluded-Duke.**

| Type | Index | Attention SEA | SPA | Rank-1 | mAP |
|---|---|---|---|---|---|
| w/o attn | 0 | ✗ | ✗ | 70.1 | 62.4 |
| w/ attn | 1 | ✓ | ✗ | 70.3 | 62.6 |
| | 2 | ✗ | ✓ | 70.5 | 62.8 |
| | 3 | ✓ | ✓ | **70.8** | **62.8** |

**Table 4: Performance of ProFD with different combination of self-distillation strategy on Occluded-Duke.**

| Type | Index | Memory global | local | Rank-1 | mAP |
|---|---|---|---|---|---|
| w/o mem | 0 | ✗ | ✗ | 68.6 | 60.2 |
| w/ mem | 1 | ✓ | ✗ | 70.7 | **62.9** |
| | 2 | ✗ | ✓ | 70.0 | 61.8 |
| | 3 | ✓ | ✓ | **70.8** | 62.8 |

DukeMTMC-ReID dataset, ProFD achieves 91.7% in rank1 accuracy and 83.2% in mAP. Clearly, ProFD demonstrates competitive performance on both of these two holistic datasets. In particular, ProFD significantly outperforms both holistic and occluded ReID methods on the DukeMTMC-ReID dataset, and achieves competitive performance on Market1501, although there still exists a slight gap compared to most state-of-the-art methods. Overall, the results above indicate that ProFD is a universal framework for preson ReID and could not compromise performance on the holistic ReID task.

### 5.4 Ablation Study

*The Effectiveness of the Hybrid Attention in ProFD.* We conducted detailed ablation studies on the Occluded-Duke dataset to assess the effectiveness of hybrid attention for the ProFD, as shown in Table 3. The baseline of our work without hybrid attention decoder is indicated in Line1, which only utilizes weighted average pooling to extract part features based on human parsing prediction. As evidenced by Line 2 and Line 3, both semantic-aware attention and spatial-aware attention individually applied to the baseline contribute to performance improvement. This improvement suggests that both attention mechanisms guided by semantic or visual information are more effective in extracting valuable information compared to simply using average pooling. Furthermore, combining the two types of attention, Line 4 achieves superior performance in their individual experiments, indicating that these two types of attention mechanisms complement each other.

*The Effectiveness of General Knowledge Preservation in ProFD.* To verify the effect of the self-distillation strategy on ProFD, we further conducted detailed comparative experiments by selecting different strategies. The experimental results are reported in Table 4. From Line 1, it can be observed that not using a memory bank leads to inferior performance because direct fine-tuning would cause CLIP to lose some generalization ability and overfit to the target dataset. Accordding to Line 2 and Line 3, it is found that using either the global memory bank or the local memory

performance on Occluded-Duke and Occluded-ReID. Our ProFD still outperforms it with improvements of +0.4% and +8.5% in mAP on Occluded-Duke and Occluded-ReID, respectively. Furthermore, on P-DukeMTMC, we also outperform previous state-of-the-art methods by a significant margin, with improvements of at least +0.7% in rank-1 accuracy and +4.9% in mAP.

Notably, Occluded-ReID, a challenging dataset characterized by occlusions, demands robust domain adaptation capabilities, because it does not provide training dataset. And our method achieves significantly better results than other methods in this dataset, which surpasses other occluded ReID methods by at least 4.7% in rank-1 accuracy and 8.5% in mAP, demonstrating that ProFD can maximize the preservation of CLIP's generalization ability.

*5.3.2 Evaluation on Holistic Person ReID Datasets.* While occluded ReID methods have primarily concentrated on addressing the specific challenge of occluded ReID, they might encounter a decline in performance in the original holistic ReID task. Thus, in this section, we also assess the proposed ProFD on the holistic ReID datasets Market1501 [21] and DukeMTMC-ReID [22]. For fair comparison, we select 9 holistic ReID methods [14, 61, 62, 65, 67−71] and 12 Occluded ReID methods [3, 4, 8−12, 15, 16, 18, 19, 72].

The results are shown in Table2. In the Market1501 dataset, ProFD gets 95.1% in rank1 accuracy and 90.0% in mAP. In the

**Table 5: Performance of ProFD with different combinations of loss functions on Occluded-Duke.**

| Index | Loss | | Rank-1 | Rank-5 | mAP |
|---|---|---|---|---|---|
| | $L_{align}$ | $L_{div}$ | | | |
| 0 | ✗ | ✗ | 68.9 | 82.1 | 61.5 |
| 1 | ✓ | ✗ | 69.9 | 82.6 | 62.2 |
| 2 | ✗ | ✓ | 70.7 | 82.3 | 62.4 |
| 3 | ✓ | ✓ | **70.8** | **83.3** | **62.8** |

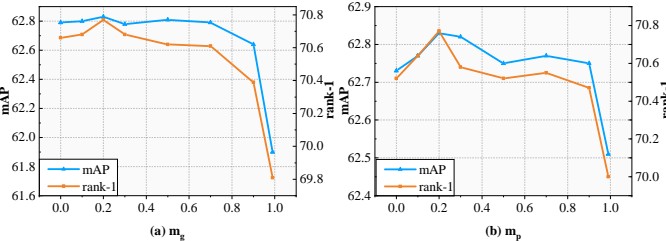

**Figure 4: Evaluation of the perfomance with different momentum $m_g$ and $m_p$ on Occluded-Duke.**

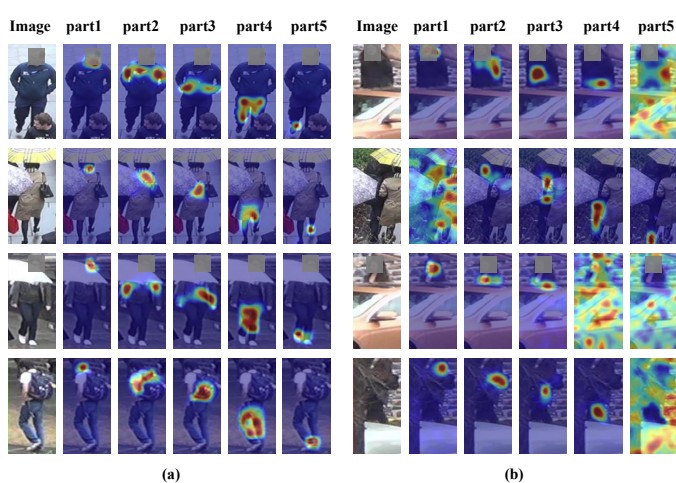

**Figure 5: Visualization of spatial-aware attention. (a) Unoccluded case. (b) Occluded case. Our method accurately focuses on the sepcified body regions following textual prompts in both cases.**

bank alone improves the model's performance, indicating that using memory bank benefits discriminativeness and representiveness of both global and features. Moreover, from Line 4, we can find that simultaneously using both memory banks can fully retain CLIP's pretraining knowledge and enhance the model's performance.

***The Combinations of Loss Function in ProFD.*** As described in section 4.1.2, the spatial-level alignment loss is responsible for restoring the locality of spatial features, aiding downstream tasks in extracting corresponding semantic local features. Meanwhile, the diverse loss aims to reduce redundancy between part features, further pushing the decoder to focus on different body regions. Therefore, comparative experiments were conducted in this part to demonstrate the performance of different combinations of loss functions. The experimental results are presented in Table 5.

From Table 5, we observe that individually adding the alignment loss contributes to improving both rank-1 accuracy and mAP slightly, as it enhances the locality of spatial features generated by CLIP. However, it does not reduce the correlation between part features. And solely adding the diverse loss can significantly enhance the model's performance, as it reduces the redundancy of local features. It provides more informative part features, which can be combined with global features to form a stronger representation. Additionally, using both loss functions together yields better performance compared to using them individually, indicating their complementary nature. The former focuses on enhancing the semantic information of part features, while the latter emphasizes enhancing the diversity of part features.

***Parameters Analysis.*** As depicted in Equation 15, the momentum value $m_g$ and $m_p$ govern the update speed of memory bank. A higher momentum value corresponds to a slower update of class center. We conducted experiments on the Occluded-Duke dataset to investigate the impact of various $m_g$ and $m_p$ values on our method. As illustrated in Figure 4, the method performs satisfactorily when

$m_g$ and $m_p$ are less than 0.9. However, when $m_g$ and $m_p$ becomes excessively large (e.g., 0.99), the accuracy significantly decreases. And compared with $m_g$, the influence of $m_p$ on the model's performance is smaller, which suggests that the global memory bank plays a more crucial role during the training process. The optimal performance is attained with $m_g = 0.2$ and $m_p = 0.2$.

## 5.5 Visualization

In addition, we present visualization of the decoder's attention in Figure 5 for qualitative analysis. Figures (a) and (b) show some examples with no/slight occlusion and severe occlusion, respectively. We can observe that our method successfully and accurately locates and focuses on the sepcified body regions following textual prompts in both cases. For some occluded regions, the attention can be observed to be scattered throughout the entire image. This is a resonable phenomenon because those regions are occluded, which indicates that our method can accurately perceive occluded parts.

## 6 Conclusion

In this paper, we propose a novel CLIP-based framework named Prompt-guided Feature Disentangling (**ProFD**), which aims to address the challenges of occluded person re-identification (ReID). To mitigate missing part appearance information caused by occlusion and noisy spatial information from external model, **ProFD** effectively generates well-aligned part features by leveraging the pre-trained knowledge of textual modality. Furthermore, to avoid the catastrophic forgetting of model, we propose a self-distillation strategy with memory banks to preserve CLIP's pre-trained knowledge. Extensive experiments on multiple datasets demonstrate that **ProFD** achieves competitive performance, establishing new state-of-the-art results in occluded person ReID. We believe that our work opens up promising avenues for further advancements in the community of occluded person re-identification.

# Acknowledgments

This work was supported by the National Science and Technology Innovation 2030 - Major Project (Grant No. 2022ZD0208800), and NSFC General Program (Grant No. 62176215).

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
