# OpenReview forum: "ProFD: Prompt-Guided Feature Disentangling for Occluded Person Re-Identification"
_acmmm.org/ACMMM/2024/Conference — MM2024 Poster_

### Official Review · Reviewer_EL7a · 2024-05-24

**Rating:** 4
**Confidence:** 4

**Summary:**

In this paper, the author attempts to explore using the vision-language model combined with the body parsing model to help the image encoder learn an occlusion-aware representation. To achieve this goal, the author proposes a Prompt-guided Feature Disentangling framework in which a hybrid-attention decoder is utilized to ensure consistency in both spatial and semantic. Meanwhile, a self-distillation strategy is also introduced to alleviate the overfitting issue,

**Strengths:**

1. The whole framework is reasonable and interesting. It is nice to see the combination of the vision language model and occluded person re-identification.

2. Ther performance in occluded-REID outperforms other state-of-the-art methods by a large margin.

**Limitations:**

1. The whole network is quite complex. It contains both a vision-language model and a body parsing model. So, the computation cost during the training of the proposed method seems very large.

2. Although adding large computations, the performance improvement on occluded-duke and O-DukeMTMC is still limited.

3. The spatial-level alignment loss aims to push the representation of the text encoder to align the body parsing result. So, why not use the body parsing result directly?

**Suitability:**

3

---

### Official Review · Reviewer_V5vc · 2024-05-24

**Rating:** 3
**Confidence:** 4

**Summary:**

This paper introduces a ProFD framework that aims to use textual prompts to disentangle the global-wise feature representation into part features. Furthermore, a self-distillation strategy is also proposed to preserve the pre-trained multi-modal knowledge and alleviate overfitting. Experimental results on both holistic and occluded person re-identification datasets demonstrate the effectiveness of the ProFD.

**Strengths:**

1. Although the entire network is complex, this paper is easy to follow.

2. Visualization results show that this framework does make sense in extracting diverse part features.

**Limitations:**

1. This work is strange. Although it uses the vision language model, it still follows the original framework to learn the body parsing results for the pre-trained body parsing model. So, why not directly use the body-parsing model? As the main contribution, the vision language model is more like a gimmick than a real important contribution.

2. The entire network is too complicated. Compared to the DPM, I cannot take advantage of this work in terms of both performance and efficiency.

**Suitability:**

3

---

### Official Review · Reviewer_bgww · 2024-05-26

**Rating:** 4
**Confidence:** 4

**Summary:**

The paper introduces ProFD, a novel framework for occluded person re-identification (ReID) that leverages pre-trained knowledge from the textual modality to generate well-aligned part features. The authors propose a Prompt-guided Feature Disentangling method to tackle challenges associated with occlusion, such as missing part appearance information and noisy spatial information from external models.

**Strengths:**

+ The paper is well-written.
+ The performance achieves state-of-the-art

**Limitations:**

- In Figures 1(a) and 1(b), the authors highlight two challenges in occluded ReID: (i) the absence of part information due to occlusion, and (ii) noise introduced by segmentation or other external models. Despite identifying these issues, the authors do not address or mitigate problem (i). Additionally, although they employ a human parsing model for spatial-level part alignment, it remains unclear how their approach avoids the noise problem typically associated with such models.

- If ordinary models cannot generalize well on ReID datasets, is fine-tuning these models on ReID-specific data available? Besides, trying a powerful and updated model like SAM may help.

- Due to privacy issue, DukeMTMC is not available anymore. A better way to study occlusion problems is to convert an ordinary dataset into occlusion one.

- The improvement of some components are limited. For instance, in Table.3, the difference of using the proposed attention mechanism or not is little, from 70.1, 62.4 to 70.8, 62.8.

**Suitability:**

3

---

### Meta-Review · Area_Chair_rAFw · 2024-07-03

**Recommendation:** Accept (Poster)
**Confidence:** 4

**Metareview:**

The paper introduces ProFD, a novel framework for occluded person re-identification (ReID) that leverages pre-trained knowledge from the textual modality to generate well-aligned part features. Pre-rebuttal this paper received diverse ratings, i.e. BA,BR,BA. The main concerns were limited improvement of some components (R1), the vision language model is more like a gimmick instead of a main contribution (R2), too complicated network (R2,R3) and limited experimental improvement (R3). Post-rebuttal the paper had consistent ratings: 2 WA, and 1 BA. The rebuttal addressed the concerns of the reviewers. AC agrees with the reviewers and tends to accept this paper.

---

### Meta-Review · Senior_Area_Chairs · 2024-07-10

**Recommendation:** Accept (Poster)
**Confidence:** 4

**Metareview:**

This paper received mixed ratings initially. After rebuttal, all the reviewers tend to accept the paper. SAC and AC agree with reviewers and recommend acceptence.